# Cardiac Autophagy in Sepsis

**DOI:** 10.3390/cells8020141

**Published:** 2019-02-10

**Authors:** Yuxiao Sun, Ying Cai, Qun S. Zang

**Affiliations:** 1Departments of Surgery, University of Texas Southwestern Medical Center, Dallas, TX 75390, USA; yuxiao.sun@utsouthwestern.edu; 2Department of Developmental Cell Biology, School of Life Sciences, China Medical University, 77 Puhe Road, Shenbei New District, Shenyang 110122, China; ycai27@cmu.edu.cn

**Keywords:** Beclin-1, autophagy, mitophagy, cardiac dysfunction, sepsis, endotoxemia

## Abstract

Sepsis is a leading cause of death in intensive care units, and cardiac dysfunction is an identified serious component of the multi-organ failure associated with this critical condition. This review summarized the current discoveries and hypotheses of how autophagy changes in the heart during sepsis and the underlying mechanisms. Recent investigations suggest that specific activation of autophagy initiation factor Beclin-1 has a potential to protect cardiac mitochondria, attenuate inflammation, and improve cardiac function in sepsis. Accordingly, pharmacological interventions targeting this pathway have a potential to become an effective approach to control sepsis outcomes. The role of autophagy during sepsis pathogenesis has been under intensive investigation in recent years. It is expected that developing therapeutic approaches with specificities targeting at autophagy regulatory factors may provide new opportunities to alleviate organ dysfunction caused by maladaptive autophagy during sepsis.

## 1. Introduction

Sepsis is a leading cause of death in intensive care units worldwide [1]. This devastating condition presents a life-threatening organ dysfunction caused by dysfunctional host response to infection [2]. Despite improvements in antibiotic therapies and critical care techniques [3], the reported incidence of sepsis is still increasing, most likely reflecting ageing populations with more comorbidities and greater recognition [4,5]. Further, patients who survive sepsis often bear long term disabilities with significant health care and social implications [6]. Sepsis has become a major public health concern. According to the Centers for Disease Control and Prevention (CDC), sepsis was accounted for over twenty billion dollars (about 5%) of total US hospital costs in 2011. It was named as the most expensive in-patient cost in US hospitals in 2014, averaging more than $18,000 per hospital stay [7]. In 2017, the CDC launched a national educational campaign, “Get Ahead of Sepsis”, which aims to raise awareness and knowledge about prevention, early recognition, and timely treatment of sepsis among the public and among healthcare providers. Though sepsis has been a subject of intensive research for decades, current management for sepsis remains to be supportive care, and most attempts of molecule-based treatments have failed clinically [8,9]. Clearly, investigation of the pathological mechanisms and exploration of new therapeutic interventions are urgently needed to advance the treatment of this devastating clinical condition.

Cardiac dysfunction is an identified serious component of the sepsis-induced multi-organ failure, and it is associated with adverse outcomes and higher mortality [10,11]. The mechanisms that underlie sepsis-induced cardiomyopathy are not well-known; exuberated inflammation, comprised metabolism, and impaired beta-adrenergic response are identified potential driving forces [12]. Because mitochondria comprise about 30% of myocardial volume [13], the pathophysiologic mechanisms of sepsis-induced cardiomyopathy via mitochondrial signaling has been a focused area of investigation. In vitro and in vivo studies using preclinical sepsis models demonstrated that sepsis triggers impairments in the structure and function of mitochondria, which abnormality leads to not only a deficiency in energy supply but also an overproduction of mitochondria-derived danger-associated molecular patterns (DAMPs), such as reactive oxygen species (mtROS) [14], fragmented mitochondrial DNA (mtDNA) [15,16,17,18], N-formyl peptides [19], cardiolipin [20], ATP [21], mitochondrial transcription factor A [22], and cytochrome c [23]. These harmful molecules exacerbate myocardial inflammation and sequential cardiomyopathy during sepsis [14,15,16,17,24].

Inflammation has been center-staged in the field of sepsis research in the past. It is now clear that sepsis is a multifaceted host response to pathogens, involving the early activation of both pro- and anti-inflammatory responses, along with major changes in nonimmunologic pathways [2]. Along this line, the sepsis-associated increase in autophagy, a lysosome-dependent process of removing damaged proteins and organelles [25], was recognized in both preclinical and clinical samples [26,27,28,29,30]. Accumulating evidence indicates that sepsis triggers autophagy in multiple organs including the heart [27,28,29]. To date, the role of autophagy in the pathogenesis of sepsis is not yet well understood. This short review is focused on a discussion of sepsis-induced changes in cardiac autophagy and the potential of targeting autophagy as a therapeutic intervention to attenuate organ dysfunction in sepsis.

## 2. Autophagy Changes Dynamically During Sepsis

Autophagy is considered to provide cellular quality control to promote survival, and is therefore adaptive, under physiological responses or mild stress. However, under severe or chronic stress, excessive or inadequate autophagy causes massive self-degradation or accumulation of toxic materials; both are maladaptive and eventually provoke cell death [31,32]. Both adaptive and maladaptive features of cardiac autophagy under the condition of septic challenge were observed in published studies. For example, investigations in vivo using a mouse cecal ligation and puncture (CLP) sepsis model [33] and in vitro using cultured, lipopolysaccharide (LPS)-challenged cardiomyocytes [34] indicated that stimulating autophagy pharmacologically protects the myocardium, thus suggesting that autophagy is an adaptive response. Conversely, reducing autophagy directly via an autophagy inhibitor or indirectly via antioxidants improved cardiac contractility in a mouse model of LPS-induced endotoxemia, and thus supporting the conclusion that autophagy is maladaptive [35]. The discrepancy of these observations may be due to the differences in experimental settings, in which the severity of sepsis, drug specificity, and timing of delivery differed.

A recent investigation suggested a status that cardiac autophagy changes dynamically in response to the severity of sepsis. The study examined the degree of autophagic responses in the heart tissue of mice challenged with LPS-induced endotoxemia and found that LPS triggered a dose-dependent autophagic response [36]. When LPS was given at low doses, modeling mild sepsis, autophagy increased proportionally to the magnitude of the insult. However, under conditions of severe sepsis, this capability declined proportionally to the severity of the insult. Further, it was detected that the failure of autophagy induction at high-dose LPS challenges correlated with the activation of mTOR signaling and the appearance of clinical signs of pathogenesis, manifested as a decline in cardiac function, elevation in inflammation and cardiac tissue injury. These observations of cardiac autophagy are consistent with previous reports in the liver showing increased autophagy in the early phases of sepsis followed by a decline near late-stage organ failure in the mouse CLP sepsis model [28,37]. Thus, these changes documented in the heart may mimic the progression of a systemic multi-organ failure during sepsis. It is speculated that, during the beginning phase of sepsis or mild sepsis, autophagy is adaptive; it provides a level of control to promote survival and sustain normal function of organs. However, with sepsis progresses to severe conditions, the autophagic responses become inadequate and therefore leads to a maladaptive outcome.

The mTOR pathway is a well-known upstream nodal point that acts to inhibit autophagy [38]. In the experimental settings described above, significant decreases in the activation status of mTOR complex were not detectable in the heart of animals challenged by low-dose LPS, compared with shams. However, the attenuation of autophagy in response to high-dose LPS is clearly associated with mTOR activation [36]. These results suggest that mTOR activation inhibits beneficial autophagic activities during severe stages of sepsis. The data also suggests that the activation of autophagy in early sepsis may be independent of mTOR. Though mTOR has long been considered a conventional regulatory switch for turning on and off autophagy, recent studies have discovered a number of autophagy stimulatory pathways whose actions do not involve mTOR. Signals in this category include the intracellular inositol-IP_3_ pathway [39], Ca^2+^-calpain pathway [40], cAMP-exchange protein activated by cAMP (Epac)-PLCε-IP_3_ pathway [40], leucine-rich repeat kinase 2 [41], and trehalose [42], etc. However, whether sepsis utilizes any of these signals or other unknown mechanisms to operate autophagy machinery in the heart as well as in other organs remains to be investigated. Future studies in this area will potentially identify new targets for the development of autophagy-based therapeutic approaches.

## 3. Beclin-1-Dependent Autophagy Protects the Heart During Sepsis

Beclin-1 is one of the first mammalian autophagy effectors identified [43,44]. This protein is ubiquitously expressed, and homozygous deletion of the *beclin-1* gene results in early embryonic lethality [45]. Beclin-1 functions as an autophagy initiation factor through interaction with PtdIns(3)-kinase (Vps34) [46]. Together, this protein complex initiates the nucleation step of autophagy to begin autophagic flux and also participates in later steps involving the fusion of autophagosomes to lysosomes [47,48,49].

Genetic mouse models with altered expression of Beclin-1 were applied to determine the role of cardiac autophagy in response to LPS-induced endotoxemia [36]. Transgenic mice with cardiac-specific overexpression of Beclin-1 and heterozygous deficiency were used to increase and decrease Beclin-1 in vivo. Forced overexpression of Beclin-1 attenuated cardiac inflammation and fibrotic injury, preserved mitochondrial quality, and ultimately improved cardiac performance in response to LPS-challenge. Additionally, the activation of mTOR was blunted in the hearts with Beclin-1 overexpression in response to high-dose LPS challenge [36]. Since the activation of mTOR is inversely correlated with autophagic activities, the result suggests that enhancing Beclin-1 signaling can suppress mTOR activation, thereby sustaining autophagy even under conditions of severe sepsis. In parallel, Beclin-1-dependent activation of AMP-activated protein kinase (AMPK) and Unc-51 like-autophagy-activating kinase 1 (ULK1) was also observed. It has been a general understanding that mTOR suppresses autophagy by inhibiting ULK1, a autophagy activating kinase that functions upstream of Beclin-1 [50]. The newly obtained data described above suggest a notion that Beclin-1 may involve a positive feedback regulation of autophagy by enhancing AMPK and ULK1 in the heart in response to the challenge by endotoxemia.

Beclin-1 mediated protection in sepsis was also suggested by a study of carbon monoxide therapy (CO) in the mouse CLP sepsis model [51]. Knockdown Beclin-1 abolished the beneficial effects of CO on attenuation of inflammation and reduction of bacteremia. Beclin-1 is also required for the CO-dependent enhancement of phagocytosis by macrophages. Thus, the protection by Beclin-1 during sepsis does not seem to be restricted to the heart. However, the regulatory function of Beclin-1 in various cell types and tissues under the pathological condition of sepsis remain further investigation.

## 4. Beclin-1-Dependent Protection of Cardiac Mitochondria

Sepsis induces mitochondrial damage, releasing mitochondrial DAMPs that aggravate myocardial inflammation and cardiac dysfunction [14,15,16,17,18,52]. During endotoxemia, LPS causes a disruption of mitochondrial structure, reduction in metabolic function, and the release of mtDNA fragments to the cytosol in the heart tissue. These manifestations of mitochondrial damage following LPS challenge were attenuated in the heart of mice with transgenic overexpression of Beclin-1, suggesting a critical function of Beclin-1 in the protection of cardiac mitochondria in sepsis [36].

As dysfunctional mitochondria can be segregated and eliminated through autophagy [25,53], a process termed mitophagy, Beclin-1-mediated quality control of cardiac mitochondria during endotoxemia is expected to be a product of an upregulated mitophagy. Indeed, overexpression of Beclin-1 leads to more formation of mitophagosomes/mitolysosomes in the heart tissue [36]. It is known that mitophagy occurs via multiple pathways. One route is regulated by PTEN-induced putative kinase 1 (PINK1) and E3 ubiquitin ligase Parkin, which target mitochondria with lost membrane potential and subsequently bring these mitochondria to autophagosomes for degradation [54,55]. Alternatively, mitophagy can be initiated via outer mitochondrial membranes (OMM)-located receptors, such as BNIP3L/NIX, BNIP3, and FUNDC1, and these receptors connect mitochondria to autophagosomes [56]. Interestingly, not all mitophagy seems to be created equal; and signals to activate mitophagy are selectively utilized in response to distinct stimuli. In the hearts of animals challenged by endotoxemia, Beclin-1 stimulates a striking difference in the spectrum of mitochondria-localized mitophagy factors, indicating a differential regulation of mitophagy pathways [36]. In response to LPS, Beclin-1 stimulates mitophagy, and this action is accompanied by a selective increase in the recruitment of PINK1 and Parkin to mitochondria, while strongly suppressing the receptor proteins BNIP3L and BNIP3. These data suggest that Beclin-1 protects mitochondria via a selective activation of a particular mitophagy pathway, instead of a bulk induction of all types of mitophagy. The data also support that PINK1-Parkin mitophagy is cardiac-protective, an adaptive response, during sepsis. Similar adaptive features of PINK1-Parkin mitophagy in the heart are also supported by previous published results [57,58].

The signal transduction pathway of whether and how Beclin-1 targets PINK1-Parkin mitophagy has not been fully understood. Several lines of investigation suggest that PINK1 recruits Parkin to damaged mitochondria via phosphorylated mitofusin 2 (Mfn2) [59,60,61]. In the experimental setting of endotoxemia described above [36], phosphorylated form of either Parkin or Mfn2 was not detectable in the heart tissue of mice with Beclin-1 overexpression using the published approach of Phos-tag Western blotting [59,60]. However, it is worthy to note that the detection of phosphorylated Parkin or Mfn2 was accomplished in cultured cells with an overexpression of protein targets [59,60]. The current method may have a limitation in detecting low contents of phosphorylation in in vivo models when PINK1, Parkin and Mfn2 are expressed at endogenous physiological levels. It is also speculated that the stimulation of PINK1-Parkin mitophagy by Beclin-1 may involve mitochondria-associated membranes (MAMs). MAMs are specialized subcellular domains that physically link mitochondria with the endoplasmic reticulum (ER) [62], and they are essential to mitochondrial health [63,64]. Beclin-1 and PINK1 are found to re-localize to MAMs during mitophagy initiation, and via direct interaction, they promote ER-mitochondria tethering and the formation of autophagosome precursors to facilitate mitophagy [65,66]. Beclin-1 was found to interact with Parkin, and this association is required for the progress of PINK1-Parkin mitophagy [67]. Thus, the PINK1-Parkin mitophagy pathway is more accessible for activation when Beclin-1 signaling is upregulated.

In addition to remove damaged mitochondria by mitophagy, Beclin-1 may promote mitochondrial biogenesis via the upregulation of PINK1/Parkin and/or AMPK/ULK1 [68,69,70,71,72,73]. Recent studies have shown that Parkin can positively regulate mitochondrial biogenesis through proteasomal degradation of its substrate PARIS, a zinc-finger protein [69,70]. In neuroblastoma cells, increases in mitochondrial and lysosomal biogenesis via transcription factor Nrf2 and TFEB were detected following PINK1-Parkin mitophagy [71]. In vivo, a lack of Parkin expression in the ventral midbrain resulted in decreases in mitochondrial size, number, and mitochondrial biogenesis makers, together with declines in mitochondrial respiration, implicating that Parkin is required for the production of new, healthy mitochondria [69]. In cultured human endothelial cells and mouse aortas, AMPK phosphorylates epigenetic factors of DNA methyltransferase DNMT1, histone acetyltransferase HAT1, and retinoblastoma binding protein 7 (RBBP7), which in turn activates the transcription of nuclear DNA encoded genes of mitochondrial biogenesis and energy production [72]. To date, whether similar mechanisms are utilized to regulate mitochondria in septic hearts remains for further investigation.

The mechanism for how Beclin-1 suppresses BNIP3 signaling also remains to be determined. Similar to the due-function of PINK1-Parkin, BNIP3 also functions as a dual regulator in mitochondria. It is regulated transcriptionally via hypoxia-inducible factor 1 [74] and FOXO3a [75] and/or by posttranslational oxidation via ROS [76]. Upon activation, BNIP3 prompts the opening of the mitochondrial permeability transition pore (mPTP), recruits LC3 for autophagosome formation, and initiates mitophagy [77,78,79]. This protein also causes mitochondrial dysfunction and subsequently induces cell death via the activation of downstream effectors Bax/Bak under stress conditions [80]. Both BINP3-induced mitophagy and mitochondrial damage were observed in the myocardium [77,81]. It was further revealed that transcriptional upregulation of BNIP3 by FOXO3a caused a decrease in mitochondrial membrane potential and an increase in mitochondrial Ca^2+^, leading to mitochondrial fragmentation and apoptosis of cardiomyocytes [75]. Hence, the effect of BNIP3 accumulation on mitochondria is a result of the balance between its double actions. In the hearts of LPS-challenged mice, the increase in mitochondrial BNIP3 may aggravate detrimental mitochondrial deficiency while the induced mitophagy is not strong enough to clear damaged mitochondria, and Beclin-1 may have a control to mitigate BNIP3 signaling [36]. BNIP3 signaling stimulates pathological responses under conditions of hypoxia [80] and ischemia/reperfusion [76], however, whether BNIP3 is pathological in sepsis-induced cardiac dysfunction and how Beclin-1 interacts with BNIP3 require further investigation.

Taken all together, the improved status of the mitochondria in the hearts by Beclin-1 is likely the result of several folds of signaling regulations. While Beclin-1 selectively facilitate PINK1-Parkin mitophagy for the clearance of damaged mitochondria, Beclin-1 also promotes mitochondrial biogenesis via PINK1/Parkin as well as AMPK/ULK1 [68,69,70,71,72,73]. These effects act together, leading to a preservation of the mass of functional mitochondria during sepsis when Beclin-1 signal is enhanced. In contrast, under septic challenge, a stronger mitochondrial BNIP3 signal and low levels of AMPK and ULK1 activation, together with the overwhelming inflammatory responses, aggravate the deterioration of cardiac mitochondria when the signal of Beclin-1 is not sufficiently available.

## 5. Autophagy/Mitophagy and Inflammation in Cardiac Dysfunction During Sepsis

Autophagy presents a control over inflammation by limiting the availability of inflammasome activators and/or components [82], and by reducing mitochondrial DAMPs via mitophagy [52]. On the other hand, an increase in autophagy under certain severe or prolonged stress may exacerbate unwanted outcomes due to deregulated degradation of cellular components, resulting in aggressive inflammation [83,84].

The influence of mitochondria and mitophagy on the regulation of inflammation has received intensive attention in recent years. It is now discovered that cytosolic mtDNA binds to intracellular TLR9 and activates inflammatory factor NF-κB [85]. Cytosolic mtDNA fragments also stimulate the DNA sensor cGAS and promote STING-IRF3-dependent pathway [86]. Furthermore, newly synthesized and oxidized mtDNA activates NLRP3-containing inflammasome by direct binding during innate immune response [87]. Therefore, mitophagy is expected to be an effective tool to remove dysfunctional mitochondria and hence reduce mitochondrial DAMPs such as mtDNA. A strong evidence comes from a recent investigation in animal models lacking mitophagy factor Parkin or PINK1. Under the stress condition of exhaustive excursive, inflammation resulted from accumulation of cytosolic mtDNA was abolished by loss of STING, supporting an important role of PINK1/Parkin mitophagy in restraining innate immunity [88].

Previous research indicates that cardiac inflammation during sepsis and trauma condition such as burn injury is mediated at least by the activation of toll-like receptor 4 (TLR4)/CD14 complex [24] and the increase in mitochondrial DAMPs [14,16]. In the model of LPS-induced endotoxemia, increasing Beclin-1-dependent autophagy improved cardiac outcomes and attenuated the production of cytokines [36]. However, this benefit comes with certain limitations, since in response to high-dose LPS (10 mg/kg), overwhelming inflammation still occurs in Beclin-1 overexpressed mice even when significant autophagic activity was provided. Thus, inflammation remains, at least in part, a major cause of cardiac dysfunction in these mice. Additionally, the possibility that the increase in autophagy itself may contribute to the heart failure can’t be excluded. Signaling crosstalk between autophagy and inflammation happens dynamically throughout the development of various disease conditions [84]. How autophagy alters individual inflammation pathways via inflammasomes, caspases, NF-κB and/or MAPKs, etc. at different stages of sepsis remains to be defined.

LPS triggers cytokine production via binding to the TLR4/CD14/MD2 receptor complex on cellular membranes and/or to caspase inside of cells, resulting in the activation of NF-κB and inflammasomes [89,90]. This action sequentially causes the release of DAMPs from activated, injured, or necrotic cells [91,92]. LPS also stimulates mtROS overproduction, leading to mitochondrial damage [93] and a release of mitochondrial DAMPs including mtDNA fragments [18], N-formyl peptides [19], mtROS [14], cardiolipin [20], ATP [21], mitochondrial transcription factor A [22], and cytochrome c [23]. Since combined signals from PAMPs and DAMPs stimulate inflammation in the heart during endotoxemia, it is not a surprise that levels of cytosolic mtDNA measured are not proportional to the production of total cytokines in the heart tissue [36]. As commented in a recent review [94], the investigation of Beclin-1-mediated cardiac protection during endotoxemia suggests that enhancing Beclin-1 signal in cardiomyocytes mitigates inflammation by suppressing the release of mitochondrial DAMPs via PINK1/Parkin mitophagy, which may in turn interrupt local inflammatory circuitries, reduce neutrophil infiltration, and inhibit further cytokine production by leukocytes and non-myocytes such as fibroblasts.

LPS triggers an aggressive accelerated overall mitochondrial deterioration that contributes to the animals’ rapid decline in cardiac performance via multiple pathways [36]. In addition to inflammation, other types of consequences of mitochondrial damage such as a deficiency in energy supply and alterations in mitochondrial metabolisms, are considered noninflammatory factors that may significantly impact cardiac dysfunction under septic challenge. Whether Beclin-1 presents regulation in these aspects of mitochondria properties remains to be determined.

## 6. Autophagy-Based Therapeutic Opportunities for Sepsis

Changes in autophagy are associated with a number of pathological disorders, ranging from inflammatory and autoimmune conditions to neurodegeneration, neuronal injury, cardiovascular diseases, cancer, hepatic and metabolic disorders. Accordingly, pharmacological modulators of autophagy are expected to provide a control over pathogenesis in disease models. Indeed, new interventions are being developed and tested at various stages [95]. However, at the current time, autophagy-based therapies have not yet been available for clinical treatments. Although certain FDA-approved drugs, such as rapamycin, chloroquine, and metformin, have effects to activate or deactivate autophagy, they were not developed for the purpose to adjust autophagy and these drugs have broad-spectrum impacts in vivo. For example, rapamycin is an inhibitor of mTOR, a serine/threonine kinase that negatively regulates autophagy [96]. mTOR is also involved in functions outside of autophagy, such as protein turnover, metabolism, energy balance, and stress responses [96]. Therefore, due to the complexity of mTOR signaling, rapamycin may cause unwanted toxicity, which is likely the reason of inconsistent test results of rapamycin in preclinical sepsis models [33,97,98,99,100]. Rapamycin was shown to improved cardiac performance [33], reduced cognitive impairments [97] in a mouse CLP-induced sepsis model. However, adverse effects of rapamycin on the survival of mice subjected to CLP sepsis [101] and on the lung injury of mice upon LPS-induced endotoxemia were also reported [99,100]. Due to the rapamycin’s lack of specificity to autophagy, it becomes difficult to draw a conclusion regarding the pathological role of autophagy in sepsis. To date, the pathways of how sepsis incites autophagic responses are not fully understood. Future success in the research of this area will greatly facilitate the development of novel pharmacological interventions that specifically target autophagy factors, which may become a much-improved approach to combat sepsis.

In recent years, emerging drug development effort has been deliciated to regulatory factors of the autophagic machinery but with limited roles in cellular processes other than autophagy. As results, new autophagy inhibitors and activators are identified. Small molecules that inhibit autophagy by targeting autophagy-related 4B cysteine peptidase (ATG4B), UNC-51-like autophagy-activating enzymes (ULK1), and LC3-associated phagocytosis (LAP) have yield promising results in in vitro studies using cultured cancer cells [102,103,104]. However, their application in specific animal disease models await further investigation. A significant leap in the development of autophagy-inducing therapy is the discovery of a cell-permeable Beclin-1 activating peptide, Tat-Beclin-1 peptide (TB peptide). TB peptide contains 18 amino acids of the BARA domain of Beclin-1, and thus induces autophagy by competitively binding of the endogenous Beclin-1 to its negative regulator, Golgi-associated plant pathogenesis-related protein 1 [105]. This peptide has demonstrated therapeutic benefits in disease models related to reducing viral infection [105], improving cardiac performance during pressure overload [106], and enhancing the effectiveness of chemotherapy [107]. Additionally, three novel small molecules with the feature of autophagy inducer have been identified by a method of high-throughput screen lately [108]. These molecules block the binding of Beclin-1 to its downstream inhibitor Bcl2 and therefore stimulate autophagic flux. Their specificity and capability to induce autophagy was confirmed in cultured cells [108].

The therapeutic potential of TB peptide in sepsis is suggested by a recent preclinical evaluation in the mouse model of LPS-induced endotoxemia [36]. TB peptide was given in mice subjected to LPS challenge. The dose of TB peptide was chosen based on published results [105,106,107] to find which dose induced sufficient autophagic flux without causing detectable toxicity in both sham and LPS-challenged mice [36]. This treatment significantly improved cardiac function and reduced circulating cytokines, especially TNF-α, IFN-γ, IL-17α, and IL-6, following LPS. Additional data showed that the peptide rescued the phonotypes in mice with heterozygous knockdown of Beclin1, including a significant improvement in survival in response to a lethal dose of LPS. The results further demonstrated that the peptide’s mechanism of action is mediated though a directly targeted stimulation of Beclin-1. Future evaluation of TB peptide in other sepsis models, such as CLP-induced sepsis and infection-induced sepsis, will provide essential evidence to estimate the therapeutic potential of this drug in sepsis. In addition, whether the effects of this peptide are tissue- or cell-specific warranties further investigation. Nonetheless, current evidence suggests that autophagy inducers such as TB-peptide, used by itself or in combination with other therapies, may holds an exciting therapeutic potential for the control of sepsis.

## 7. Summary

Autophagy plays an important role in sepsis-induced pathology of organ dysfunction. Accumulating evidence indicates that autophagy dynamically changes during the progress of sepsis, with insufficient and maladaptive autophagy occurs at the later stage of sepsis. This deficiency is associated with an upregulation of mTOR signaling, and the ineffectiveness of removing dysfunctional organelles and toxic materials intracellularly results in an overwhelming accumulation of DAMPs. Specific targeting autophagy factors offers an opportunity of developing novel therapeutic interventions for sepsis. For example, a newly developed cell permeable peptide, TB peptide that activates Beclin-1, showed therapeutic promise in a mouse model of LPS-induced endotoxemia [36]. Important questions await to be addressed, such as cell type or tissue-specific autophagic responses and their changes during the course of sepsis development. Additionally, evaluation of a potential therapy needs to be further validated in multiple preclinical models. Future investigations to answer these questions and new discoveries of sepsis-associated pathways of autophagy will provide a great potential to identify additional therapeutic targets for sepsis.

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
