# Peer review of "Cardiac Autophagy in Sepsis"

_cells, 2019, doi:10.3390/cells8020141_

Reviewer 1 Report

The manuscript under review by Sun et. al is a review report of current knowledge regarding Cardiac Autophagy in Sepsis. The review is very vivid as the authors compare and contrast different studies and provide the deeper insight. They also provide very thoughtful future direction. Overall the manuscript is well written and organized.

Author Response

This reviewer commented that the manuscript was well organized and written. 

Reviewer 2 Report

The review "Cardiac autophagy in sepsis" by Sun et al. discusses the recent literature concerning the implication of autophagy in the development of sepsis, with an emphasis on Beclin-1-dependent signalling.

Major point:

- Due to the complexity of the signallings and processes referred to in the text, some diagrams would be helpful to readers who are not familiar with them.

Minor points:

l.47: the full term for DAMPs should be used before the abbreviation (as in l.50)

l.69: full term for LPS

l.124-125: the sentence starting with "Because mTOR suppresses autophagy [...]" is not clear, please rephrase.

l.162-163: the sentence is not clear, please rephrase.

l.2017: IBNIP3 -> BNIP3

l.227: NFkB (kappa character does not appear properly)

l.268: "et al." is not suitable for a list, consider replacing it.

l.294: "in vitro" to be in italic font.

Author Response

While this review was under written, the authors' study of Beclin-1-mediated cardiac protection in response to LPS was commented in a review by Dr. Kroemer, published in Ann Trans Med (Nov. 2018). The hypothesized signal pathways were well summarized and diagramed in this review. In this revision, this comment review article was added to the reference, and a paragraph was added (and underlined) accordingly to describe the potential pathways.

Other minor errors were corrected according to the reviewer's comments.

Reviewer 3 Report

The manuscript titled “cardiac autophagy in sepsis”from Sun Y et al made a summary of the changes on autophage in heart during sepsis pathogenesis and its regulating signaling pathway involved. Specifically, the author demonstrated the important role of Beclin-1 on sustain autophagy in heart during sepsis and protect cardiac mitochondria via several signaling regulations.  Finally, the authors introduced the progress on drug discovery by autophagy activation or inactivation for clinical treatments especially in sepsis model.

Overall, the manuscript here is well-written and the summary is clearly presented. I recommend its publication in Cells. The manuscript could be further improved by introducing 1-2 graphic to vividly picture the signaling pathways importantly involved in autophagy as well as its critical regulators.

Author Response

As addressed previously to reviewer #2, the authors' study of Beclin-1-mediated cardiac protection in response to LPS was just commented in a review by Dr. Kroemer, published in Ann Trans Med (Nov. 2018). The hypothesized signal pathways were well summarized and diagramed in this review. In this revision, this comment review article was added to the reference (ref #94), and a paragraph was added (and underlined) accordingly to describe the potential pathways.